# miRNome and Proteome Profiling of Small Extracellular Vesicles Secreted by Human Glioblastoma Cell Lines and Primary Cancer Stem Cells

**DOI:** 10.3390/biomedicines10081886

**Published:** 2022-08-04

**Authors:** Ingrid Cifola, Federica Fratini, Beatrice Cardinali, Valentina Palmieri, Giuliana Gatti, Tommaso Selmi, Sara Donzelli, Andrea Sacconi, Valeriana Cesarini, Hany E. Marei, Massimilano Papi, Giovanni Blandino, Carlo Cenciarelli, Germana Falcone, Igea D’Agnano

**Affiliations:** 1Institute for Biomedical Technologies (ITB), CNR, 20054 Segrate, Italy; 2Proteomics Core Facility, Istituto Superiore di Sanità (ISS), 00161 Rome, Italy; 3Institute of Biochemistry and Cell Biology (IBBC), CNR, 00015 Monterotondo, Italy; 4Fondazione Policlinico Universitario A. Gemelli IRCSS, 00168 Rome, Italy; 5Dipartimento di Neuroscienze, Sez. Fisica, Università Cattolica del Sacro Cuore, 00168 Rome, Italy; 6Institute for Complex Systems (ISC), CNR, 00185 Rome, Italy; 7Department of Biotechnology and Translational Medicine, University of Milan, 20129 Milan, Italy; 8Translational Oncology Research Unit, IRCCS Regina Elena National Cancer Institute, 00144 Rome, Italy; 9Clinical Trial Center, Biostatistics and Bioinformatics Unit, IRCCS Regina Elena National Cancer Institute, 00144 Rome, Italy; 10Institute of Translational Pharmacology (IFT), CNR, 00133 Rome, Italy; 11Department of Cytology and Histology, Faculty of Veterinary Medicine, Mansoura University, Mansoura 35116, Egypt

**Keywords:** glioblastoma, cancer stem cells, extracellular vesicles, miRNAs, proteome

## Abstract

Glioblastoma (GBM) is the most common and aggressive brain tumor in adults. Despite available therapeutic interventions, it is very difficult to treat, and a cure is not yet available. The intra-tumoral GBM heterogeneity is a crucial factor contributing to poor clinical outcomes. GBM derives from a small heterogeneous population of cancer stem cells (CSCs). In cancer tissue, CSCs are concentrated within the so-called niches, where they progress from a slowly proliferating phase. CSCs, as most tumor cells, release extracellular vesicles (EVs) into the surrounding microenvironment. To explore the role of EVs in CSCs and GBM tumor cells, we investigated the miRNA and protein content of the small EVs (sEVs) secreted by two GBM-established cell lines and by GBM primary CSCs using omics analysis. Our data indicate that GBM-sEVs are selectively enriched for miRNAs that are known to display tumor suppressor activity, while their protein cargo is enriched for oncoproteins and tumor-associated proteins. Conversely, among the most up-regulated miRNAs in CSC-sEVs, we also found pro-tumor miRNAs and proteins related to stemness, cell proliferation, and apoptosis. Collectively, our findings support the hypothesis that sEVs selectively incorporate different miRNAs and proteins belonging both to fundamental processes (e.g., cell proliferation, cell death, stemness) as well as to more specialized ones (e.g., EMT, membrane docking, cell junction organization, ncRNA processing).

## 1. Introduction

Glioblastoma multiforme (GBM) is the most common and malignant brain tumor. GBM is a high-grade glioma with an annual incidence of 4–5 cases per 100,000 people in Europe and a median onset age of 63 years. The median survival of GBM patients is 14–15 months from clinical diagnosis [1]. Despite multiple available therapeutic modalities, GBM remains a devastating fatal tumor. The intra-tumoral heterogeneity of GBMs is a major factor contributing to the poor clinical outcomes associated with these highly aggressive glial tumors and a major obstacle to effective treatments [2]. The bulk of cells that compose a GBM is thought to derive from a small heterogeneous population of cancer stem cells (CSCs) [3,4]. CSCs, or tumor-initiating cells, are a subset of tumor cells that possess the stem cell properties of self-renewal and multi-lineage differentiation and are highly efficient at initiating tumor xenografts in vivo. Such cells are proposed to persist in recurrent GBM and to cause relapses after standard therapies [5].

It has recently been proposed that in GBM, most cells in the bulk can adopt the features of CSCs rather than being restrained in hierarchically organized clonal populations. According to this hypothesis, phenotypic GBM heterogeneity is the result of reversible state transitions influenced by the tumor microenvironment (TME) and provides a growth advantage in vivo [6].

Researchers have increasingly focused on GBM intra-tumor heterogeneity, looking at how GBM cells interact with the various cell populations in the TME to learn more about tumor growth, invasion, and treatment responses. The TME is made up of tumor cells, CSCs, non-immune (fibroblasts, endothelial cells) and immune cells. GBM is characterized by the absence of T-cell infiltration [7], recruitment of immunosuppressive cells (e.g., tumor-associated macrophages), and release of tumor-derived immunosuppressive factors, all of which are obstacles to effectively treating this tumor [8,9]. An emerging mode of intercellular communication between GBM and the surrounding microenvironment is through the release of tumor-derived extracellular vesicles (EVs). EVs are membranous particles released by all cell types and include both small (40–100 nm) endocytic nanovesicles (small EVs, also called exosomes) and larger (50–1000 nm) vesicles derived via plasma membrane budding (called microvesicles) [10]. EVs contain nucleic acids, proteins, lipids, and metabolites capable of activating signaling pathways, silencing target genes, and inducing the translation of effector proteins in recipient cells [11,12]. GBM cells secrete many more EVs than normal cells, and the composition of these EVs may make the TME more permissive to tumor spreading [13,14]. Skog et al. [13] discovered that EVs produced by GBM cells can cross the blood–brain barrier (BBB) and transfer their molecular cargo into peripheral circulation, contributing to tumor dissemination and heterogeneity.

Among EV cargo, miRNAs have been particularly studied for their ability to regulate gene expression even at a distance [15]. EV-miRNA signature is often significantly distinct from that of its cell of origin, and it changes under pathological conditions, suggesting that miRNAs are selected prior to being packaged into EVs [13,16]. miRNAs contained in EVs released from GBM cells have been shown to have both pro-tumor and anti-tumor activities. EV-packaged miR-30b-3p released from hypoxic GBM stem-like cells conferred temozolomide resistance [17]. Nonetheless, several studies have revealed that EV-mediated miR-124 treatment has anti-tumorigenic properties, reducing M2 microglial polarization and decreasing GBM cell proliferation [18]. These data imply that EV-derived miRNAs could be exploited as diagnostic, prognostic, and monitoring biomarkers for GBM [19,20].

Along with EV-associated miRNAs, several proteomic profiling studies of EVs have been performed to identify disease-associated proteins useful as prognostic and/or diagnostic biomarkers [21] and to investigate the role of the EV proteome in biological phenomena, such as disease progression [22]. Several studies trying to select for potential cancer biomarkers have identified arrays of protein cargo of GBM-derived EVs [23,24], and a database of proteoforms for GBM as well as for normal fibroblasts was developed [25]. Interestingly, it was shown that many proteins with enhanced expression levels in GBM cells are present also in the small EVs released from these cells [24].

Numerous studies have been conducted to identify miRNA and protein signatures of GBM-derived EVs. However, due to significant variations between these investigations, these biomarkers have yet to be implemented in clinical practice. Combining miRNA and protein profiling from the same EVs may expand biomarker discovery options and increase our understanding of their biological roles, assisting in the identification of the most relevant pathways involved in GBM progression.

In the present study, we investigated the miRNome and proteome of the small EVs (sEVs) released by two different human GBM cell lines and by GBM primary cancer stem cells (CSCs) derived from neurospheres obtained from three different patient tumors. Our aim was to describe the content of sEVs in functionally distinct different tumor backgrounds and to assess whether they may target different cellular pathways that converge on the same biological functions. Our data indicate that sEVs derived from GBM-lines carry miRNAs with tumor suppressor function, while oncoproteins are significantly represented in their protein cargo. Conversely, in CSC-derived sEVs, we found miRNAs with established pro-tumor activity and proteins related to glioma stemness, cell proliferation, and apoptotic pathways. Collectively, these findings support the hypothesis that sEVs of different origin incorporate selected miRNAs and regulatory proteins belonging both to fundamental processes (e.g., cell proliferation, cell death, stemness) as well as to more specialized ones (e.g., epithelial-to-mesenchymal transition (EMT), membrane docking, cell junction organization, ncRNA processing).

## 2. Materials and Methods

### 2.1. Cell Line Cultures

The original U373 MG-Uppsala (U373) human glioblastoma cell line was purchased from the ECACC (European Collection of Authenticated Cell Culture, Porton Down, Salisbury, UK) and cultured in EMEM medium supplemented with 10% fetal calf serum (FCS, Hyclone, Logan, UT, USA), L-glutamine (2 mM), non-essential amino acids (NEAA, 1%), sodium pyruvate (1 mM), and antibiotics (penicillin/streptamicin) at 37 °C and 5% CO_2_ in a humidified incubator.

The U87 MG (U87) was a kind gift from Regina Elena Institute (Dr. Carlo Leonetti, National Institute of Tumors Regina Elena, Rome, Italy). The genetic identity of the U87 MG cell line was confirmed using PCR-single-locus technology (Genetica, Palo Alto, CA, USA). The short tandem repeat (STR) analysis of the microsatellite motifs was performed using the PowerPlex 16HS System (Promega, Milan, Italy). The PCR products were run in the ABI 3130XL capillary electrophoresis (Thermo Fisher Scientific, Waltham, MA, USA), and the electropherograms were analyzed using the Gene Mapper ID-X software (Thermo Fisher Scientific). The 16 loci analyzed were: D5S818, D7S820, D8S1179, D13S317, D16S539, D18S51, CSF1PO, Penta D, TH01, D21S11, Amelogenin, Penta E, vWA, TPOX, and FGA, plus a mouse marker to detect any cross-contamination with mouse DNA. Loci analysis was performed by the Cancer Research UK Cambridge Institute, University of Cambridge. U87 MG cells were cultured in DMEM medium supplemented with 10% FCS (Hyclone), L-glutamine (2 mM), and antibiotics (Penicillin/Streptamicin) at 37 °C and 5% CO_2_ in a humidified incubator.

### 2.2. Cancer Stem Cells

Procedures for collection of adult human GBM CSCs were approved by the Ethical Committee of the Catholic University of Rome, as previously reported [5]. The ethical principles of the Declaration of Helsinki were strictly followed.

CSC cells were retrieved from three adult patients affected with GBM and undergoing craniotomy at the Institute of Neurosurgery, Catholic University School of Medicine of Rome, Italy. Cells grow spontaneously in suspension as neurospheres in the presence of human recombinant EGF (20 ng/mL; PeproTech, Rocky Hill, NJ, USA) and human recombinant bFGF (10 ng/mL; PeproTech) in serum-free medium DMEM/F12 (1:1) (Invitrogen, Thermo Fisher), as previously described. Mid-sized neurospheres were enzymatically dissociated using Accutase (Merck Millipore, Darmstadt, Germany) for 1–2 min at 37 °C and replated as single cells for healthy cell proliferation. In order to have good preparations in terms of abundance and integrity of sEVs, CSCs from each patient were plated on matrigel-pre-coated 175 cm^2^ flasks to achieve 80–90 percent confluence by 48 h. Before moving on to the sEV purification process, the supernatants were pooled together.

### 2.3. sEV Isolation and Purification

U373 and U87 cell lines were seeded in growth medium for 48 h to obtain 80 percent confluence. FCS was depleted from endogenous EVs prior to use by ultracentrifugation at 100,000× *g* for at least 12 h, using an L8-70MK Ultracentrifuge (Beckmann Coulter, Pasadena, CA, USA). After centrifugation, the FCS supernatants were filtered with a 0.22 µm filter (Thermo Fisher Scientific) and stored in aliquots at −80 °C. Cell plates were then washed twice with PBS (Gibco, Thermo Fisher) and incubated for 24–36 h in medium supplemented with 10% EV-free FCS.

Differential centrifugation was used to separate sEVs from U373, U87, and CSC cell culture medium. To eliminate dead cells and apoptotic bodies, a first centrifugation at 300× *g* for 10 min was followed by a second at 2000× *g* for 30 min. To eliminate any remaining cell organelles or large vesicles, a third centrifugation at 20,000× *g* was performed. sEVs were then pelleted from the purified supernatants by ultracentrifugation at 100,000× *g* for 70 min in 38 mL polycarbonate tubes (Beckman). These sEV-enriched pellets were resuspended in PBS and ultracentrifuged again as above, prior to be used for subsequent applications.

### 2.4. Dynamic Light Scattering (DLS)

All measurements were made at 25 °C using Zetasizer Nano ZS spectrometer (Malvern, Worcestershire, UK) equipped with a 5 mW HeNe laser (wavelength λ = 632.8 nm) and a non-invasive back-scattering optical setup. Solvent-resistant micro cuvettes (ZEN0040, Malvern, Herrenberg, Germany) were used for experiments with a sample volume of 40 μL. Results are given as mean ± standard deviation of ten replicates. sEV size distribution and concentration were calculated by DLS-based non-invasive tool. Briefly, DLS was used to measure the intensity distribution of each sample [26]. sEVs were approximated to core-shell spherical lipid bilayer vesicles to obtain a form factor. The refractive index of the sEV shell was set equal to that of the plasma membrane (nL = 1.46), and the refractive index of cytoplasm (nC = 1.38) was used for the sEV core. The Rayleigh ratio (RR) of the samples was calculated by obtaining the intensity of light scattered by each sample and by the buffer solution. When the area of the intensity distribution obtained with DLS is set equal to the RR and the number distribution of the sample is calculated using the form factor, the concentration of sEVs per ml can be calculated. More details can be found in Palmieri et al. [26].

### 2.5. Flow Cytometry

FACS characterization of sEVs purified from cell culture media was performed as previously described [27]. Briefly, after immunocapturing the vesicles with magnetic beads of 4 µm diameter conjugated with the anti-CD63 tetraspanin antibody (Dynabeads, Invitrogen, Thermo Fisher Scientific), the bead-bound sEVs were subjected to indirect or direct immunofluorescence to detect the presence of specific surface markers. Antibodies used were anti-hCD81-PE (R&D Systems, Minneapolis, MN, USA), anti-hCD9-FITC (R&D Systems, clone #209306), anti-ZONAB (Sigma-Aldrich, St. Louis, MO, USA), anti-GFAP (DakoCytomation, Glostrup, Denmark). Samples were measured using a FACSCalibur cytofluorimeter (BD Biosciences, Franklin Lakes, NJ, USA). Each experiment was performed in triplicate.

### 2.6. Total RNA Extraction from sEVs

Total RNA was extracted from the pelleted sEVs purified from U373, U87 and CSC cell culture medium using the Fatty Tissue RNA Purification Kit (Norgen Biotek, Thorold, ON, Canada) according to the manufacturer’s protocol. Before RNA extraction, the sEV preparations were treated with proteinase K followed by RNAse A to avoid contamination of the internal sEV cargo by RNA adsorbed to the vesicle external surface. The isolated RNA was accurately quantified using the highly sensitive Quant-iT RiboGreen RNA Assay (Thermo Fisher Scientific) and quality-checked on Agilent 2100 Bioanalyzer using the Agilent RNA 6000 Pico kit (Agilent Technologies, Santa Clara, CA, USA).

### 2.7. sEV-miRNA Microarray Expression Profiling

Total RNA samples extracted from purified sEVs were processed for miRNA expression profiling on Agilent microarray platform. Briefly, 100 ng of total RNA per sample was labeled and hybridized on the Human miRNA Microarray (Release 21, Agilent Technologies) containing probes for 2549 human miRNAs from the Sanger database, according to manufacturer’s protocol. Each sample was analyzed in duplicate.

Scanning and image analysis were performed using the Agilent DNA Microarray Scanner (P/N G2565BA, Agilent Technologies). Feature Extraction Software (Version 10.5, Agilent Technologies) was used for data extraction from raw microarray image files using the microRNA_105_Dec08 FE protocol. Array data are available in the Gene Expression Omnibus (GEO) repository under accession number GSE206675. 

Raw data were quantile-normalized and filtered to keep only probes detected in both replicates of at least one sample using the AgiMicroRNA Bioconductor package (v.2.34.0) [28] in R environment (v.3.6.0). Differential expression comparisons were performed using LIMMA in the AgiMicroRNA R package, with multiple testing correction via the Bonferroni–Hochberg (BH) method. A |log2FC| > 2 and BH-adjusted *p*-value < 0.01 were used as thresholds to define statistically significant differentially expressed miRNAs.

### 2.8. Bioinformatics Analysis of miRNA Target Genes

Target genes for miRNAs of interest were retrieved from the miRTarBase database (v.8.0), containing experimentally validated miRNA–target interactions (https://mirtarbase.cuhk.edu.cn/, accessed on 28 October 2020) [29]. All types of validation assays were considered.

To reduce the size of target gene lists, we kept only those genes reported as expressed in normal human brains in the GTEx Portal (https://gtexportal.org/, accessed on 8 February 2021) [30]. Target genes were maintained if they were found expressed (>0.1 TPM) in at least one of the three brain tissues—cortex (205 cases), frontal cortex (175 cases), or substantia nigra (114 cases)—included in the RNAseq normalized gene count file downloaded from the GTEx database (GTEx_Analysis_2017-06-05_v8_RNASeQCv1.1.9_gene_median_tpm.gct.gz).

ToppFun tool was used to perform functional enrichment analysis for Gene Ontology categories (https://toppgene.cchmc.org/, accessed on 9 April 2021). An FDR (BH correction) < 0.01 was applied to all the annotation terms to define statistically significant enrichments. For selected functions of interest, related GO Biological Process (GO-BP)-enriched terms were selected from ToppFun results and inspected for term hierarchy using the QuickGO webtool (https://www.ebi.ac.uk/QuickGO/, accessed on 12 April 2021); then, the genes enriching GO-BP were summed to obtain the list of unique target genes involved in each biological function of interest, and the minimum FDR was recorded. A bubble plot was created using the ggplot2 R package.

### 2.9. sEV Proteomics by Mass Spectrometry

Three biological replicates of sEVs released by the same number of cells (10^8^ cells/preparation) were prepared for each cell line to quantitatively compare the sEV-secreted proteomes. Proteins were extracted from purified sEVs by resuspending the pelleted vesicles in slightly modified Laemmli sample buffer (60 mM Tris-Cl pH6.8, 2% sodium dodecyl sulphate, 10% glycerol, 0.01% bromophenol blue, 0.1% sodium deoxycholate, and 50 mM TCEP reducing agent) at 65 °C for 45 min. PAGE fractionation and digestion was performed as already described [31]. Each biological replicate was run in six fractions. Nano-RPLC was performed using a nano-UHPLC 3000 Ultimate (Dionex; Thermo Fisher Scientific) connected to a Tribrid Orbitrap Fusion (Thermo Fisher Scientific). Tryptic digests were first trapped on a C18 RP-precolumn (5 mm × 300 µm id; LC Packings-Dionex; Thermo Fisher Scientific) and then run on a home-packed 20 cm × 75 µm id fused-silica column (8 PicoTip Emitter, New Objective, Littleton, MA, USA) packed with ReproSil-Pul C18-AQ 1.9 um beads (Dr. Maisch GmbH, Ammerbuch, Germany) for chromatographic separation. Peptides were eluted at 0.25 µL/min along a 60 min linear gradient from 5% to 35% of buffer B (95% acetonitrile, 0.1% formic acid). Full-scan mass spectrometry (MS) was acquired in Orbitrap at 60K resolution—maximum injection time of 50 ms, 1 microscans, wide quadrupole isolation activated in a mass range of 350–1550, and an AGC target of 4E5. The MS/MS scans were automatically acquired in the ion trap for a total cycle time of 3 s; quadrupole isolation window 1.6; minimum intensity 5E3; HCD fragmentation; NGC 32; normal scan rate; maximum injection time of 35 ms; AGC 5E3. Dynamic exclusion allowed a repeat count of 1 within 45 s; max tolerance 10 ppm.

### 2.10. Proteomics Data Analysis

Raw files were analyzed with MaxQuant software (v. 2.0.3.0, Max Planck Institute of Biochemistry, Munich, Germany) [32], using the SwissProt-reviewed human database (2020) and including contaminants. Protein and peptide FDR was set to 0.001, a minimum of one unique peptide was set for protein identification, while a minimum of two unmodified peptides (unique and razor) was set for protein quantification. Peptides were used only once for protein quantification (razor peptide fashion). Carboamidomethylation of cysteins was set as fixed modification, with methionine oxidation and protein N-term acetylation as variable modifications. Raw files, MaxQuant parameters, and result files are available at MassIVe and ProteomeXchange repositories (MSV000088671; PXD030894). The obtained matrix (4295 items) was further processed with Perseus software (v.1.6.15.0) [33]. Proteins identified only by site or in the decoy database or potential contaminants were removed, and a valid value in at least two biological replicates of a sample group was set, reducing the final dataset to 2445 identified and quantified proteins. The values of the three replicates were averaged by the median calculation and the ANOVA statistical multiple-sample test (FDR < 0.05) was run to identify the proteins that were significantly different between the three samples, and *t*-tests (FDR < 0.05) were run to identify protein differences in each pair of samples. To focus on differences between primary CSCs and established GBM cell lines, the latter were averaged, and a *t*-test (FDR < 0.05) was run to obtain the volcano plot of the CSC- vs. U-lines datasets. For hierarchical clustering, missing values were replaced from normal distribution and the Z-score normalization was performed. The Euclidean distance heat map was divided into nine clusters with a threshold distance of 4.5.

For protein enrichments and network graphing, the Cytoscape open-source platform (v. 3.9.0) and the String Application found herein were used [34].

## 3. Results

### 3.1. GBM Established Cell Lines and GBM Primary CSCs Secrete Small EVs Presenting Similar Surface Markers

Since sEVs are the most studied vesicle type in the crosstalk among cancer cells, CSCs and the surrounding tumor microenvironment, we investigated the sEV secretion pattern from three primary GBM CSC cultures as well as from two GBM-established cell lines, namely U87 and U373. To characterize secreted sEVs, we first measured their size using DLS, and then we assessed the expression of some surface markers by FACS. According to DLS, the mean size of released sEVs did not differ among the samples (Figure 1A). Moreover, all GBM- and CSC-derived sEVs presented high levels of the known EV surface markers CD81 and CD9 tetraspanins. In addition, in all sEV preparations, we detected the cell membrane neural marker Glial Fibrillary Acidic Protein (GFAP) and the tight junction component ZO-1–associated nucleic-acid-binding protein (ZONAB) on the sEV surface (Figure 1B).

### 3.2. Expression Profiling and Target Gene Analysis of miRNAs Contained in sEVs Secreted by GBM Cell Lines and Primary CSCs

sEVs purified from U87, U373 and CSC cell culture media were profiled for miRNA expression using microarray technology. Since CSCs released low amounts of sEVs due to their relatively slow proliferation rate, the sEVs secreted by the three CSC cell lines were pooled together and analyzed as a whole. After raw data normalization and filtering, we found a total of 593, 321, and 389 miRNAs expressed in the sEVs secreted by U87, U373, and CSCs, respectively (Appendix A). The three samples showed different overlaps, with 195 miRNAs common to all the samples (Figure 2A).

When examining the levels of miRNA expression, the top 10 most abundant miRNAs in each sample mostly overlapped, indicating that the three samples had similar percentages of expression (Figure 2B). Regardless, the unsupervised hierarchical clustering of the first 50 most variable miRNAs showed that U87- and U373-sEVs clustered together and apart from primary CSC-sEVs (Appendix A).

Next, we performed differential expression analysis to identify miRNAs that distinguish sEVs derived from primary CSCs from those released by GBM-established cell lines, and we found 147 and 226 miRNAs that were differentially expressed in CSC- vs. U87-sEVs and in CSC- vs. U373-sEVs, respectively (Appendix A). Then, we focused our attention on the top 10 most up-regulated (top10) miRNAs in GBM- as compared to CSC-sEVs and on the top 10 miRNAs in CSC- as compared to GBM-sEVs (Table 1). Since the top 10 miRNAs most up-regulated in each cell line-sEVs as compared to CSC-sEVs were identical, in the subsequent analyses, we considered the two cell lines U87 and U373 as a whole.

Concerning the top 10 miRNAs of GBM-sEVs, all were miRNAs exclusively expressed in cell line sEVs and were absent in CSC-sEVs; inversely, the top 10 miRNAs in CSC-sEVs were expressed only in CSC-sEVs and were absent in GBM-sEVs (except for miR-363-3p, found at low levels also in U373-sEVs). This might suggest that these miRNAs distinguish the two sample types and may possibly target separate functions. When we classified these top 10 miRNAs according to their tumor activity as reported in published literature, we surprisingly observed that the top 10 most up-regulated miRNAs in GBM-sEVs all have tumor suppressor activity, while the top 10 miRNAs in CSC-sEVs included three miRNAs with known pro-tumor activity and three others with tumor suppressor functions. We also discovered miRNAs in both samples with not-described cancer-related activity.

For the two sets of top 10 miRNAs, we identified their target genes in the miRTarBase database, which contains only experimentally validated miRNA–target interactions. Globally, we retrieved 1188 and 938 validated target genes for the top 10 miRNAs in GBM- and CSC-sEVs, respectively. To reduce target gene lists, we kept only those genes reported in the GTEx Portal as being expressed in normal human brain, which is the tissue anatomically surrounding GBM and receiving its secreted vesicles. After this filter, we maintained 1136 and 880 target genes expressed in brain parenchyma for GBM- and CSC-sEVs, respectively (Appendix A).

To explore the biological functions targeted by our two sets of selected miRNAs distinguishing primary CSCs and GBM cell lines, we performed functional enrichment analysis on the 1136 and 880 target genes identified for GBM- and CSC-sEVs, respectively (Appendix A). In contrast to miRNAs in CSC-sEVs, we discovered that the target genes of miRNAs in GBM-sEVs were substantially more enriched in different biological functions (Figure 3 and Appendix A). In fact, while cell proliferation, cell death and stemness functions were similarly represented in both GBM cell lines and primary CSCs, target genes of miRNAs in GBM-sEVs were also involved in other biological functions very relevant for tumorigenesis, such as migration, adhesion, EMT, and angiogenesis (Figure 3 and Appendix A).

Of note, the analysis of the main target gene functions confirmed that sEVs secreted by the GBM cell lines were enriched with tumor suppressor miRNAs, while sEVs derived from GBM primary CSCs carried many miRNAs with oncogenic functions (see Table 1).

### 3.3. Proteome Profiling of sEVs Secreted by GBM Cell Lines and Primary CSCs

sEVs purified from U87, U373, and CSC cell culture media were analyzed for proteome characterization using mass spectrometry. After raw data processing, we identified and quantified, with at least two peptides in at least two biological replicates per sample, a total of 2445 proteins: 1066, 1606, and 1989 proteins were carried in the sEVs secreted by U87, U373, and CSC cells, respectively, with 788 proteins common to all three samples (Figure 4 and Appendix A). Interestingly, 1099 of the identified proteins have already been reported in previous proteomics studies of sEVs derived from glioblastoma cells performed by mass spectrometry and submitted to Vesiclepedia 4.1 (http://microvesicles.org/, accessed on 29 November 2021) [24,35,36,37] (Appendix A).

According to the MISEV 2018 guidelines [10] established by the International Society for Extracellular Vesicles to assess the quality of EV preparations, we annotated 416 proteins (17%) in the defined main MISEV categories, of which only 21% are classified as components of non-EV co-isolated structures (Figure 5).

Moreover, we annotated a total of 93 proteins included in the ExoCarta Top100 reports (http://www.exocarta.org/, accessed on 29 November 2021), 88 of which were identified in all three samples (Appendix A). Matching the MISEV guidelines and Exocarta Top100 reports, we confirmed the good quality of our sEV preparations and assessed the presence in all three samples of 36 proteins that could be used as sEV markers, and particularly as GBM-sEV markers. These include transmembrane proteins and tetraspanins (CD81, CD9, GNA family G-proteins, integrins, Rab7, Tfr1, Bsg) as well as cytosolic proteins (TSG101, Flot-1, Annexins, Alix, Syntenin-1, RhoA, HSPA8, HSP90s) (Table 2).

A total of 534 proteins with significant quantitative differences between the three samples were identified: 176 in CSC- vs. U87-sEVs, 275 in CSC- vs. U373-sEVs, and 163 in U87- vs. U373-sEVs (Appendix A). Focusing on proteins quantitatively different between GBM- and CSC-sEVs, we find a total of 249 proteins (Appendix A). Among them, we found some important GBM markers and interesting signatures of GBM primary CSCs. Particularly, among the proteins over-expressed in GBM-sEVs, we noted the glioma-associated extracellular matrix antigen Tenascin C (TNC), the stemness-associated CD109 antigen, the tumor-associated CD44 antigen, the Ras suppressor protein 1 (RSU1). Meanwhile, among those over-expressed in CSC-sEVs, we found proteins related to glioma stemness, such as MAP4K4, N-Cadherin (CADH2), Copine III (CPNE3), and Ephrin receptor type 2 (EPHA2) (highlighted in purple in Appendix A). When considering the proteins specifically identified in GBM-sEVs, we found some popular onco-related proteins, such as AKT1, AKT2, MTOR, MAP2K1, MAP2K2, and MAPK3 (highlighted in red in Appendix A). Among those specifically identified in CSC-sEVs, we found proteins related to differentiation, proliferation, and apoptotic pathways, such as CDK2, CDK6, MSH2, MSH6, IGF1R, NOTCH1, and TRAF4 (highlighted in light blue in Appendix A). In addition, among the proteins expressed in GBM- and CSC-sEVs, we noted 10 proteins, namely ADAR, ELP1, FBRL, HCD2, METTL1, NAT10, PUS7, TRM112, TRM6, and TRMT61A, which are known RNA modifying enzymes regulating gene expression (Table 3 and highlighted in light green in Appendix A). Interestingly, seven out of these 10 proteins were uniquely expressed in CSC-sEVs (ADAR, METTL1, NAT10, PUS7, TRM112, TRM6, and TRMT61A), while the other three were also expressed in U373-sEVs (FBRL and HCD2) or only expressed in U373-sEVs (ELP1).

### 3.4. Functional Analysis of Proteins Contained in sEVs Secreted by GBM Cell Lines and Primary CSCs

To explore the biological role of the proteome cargo of our sEVs, we performed functional enrichment analysis of the global dataset of proteins identified in our samples, as a whole and divided for cell line, and found that several biological functions crucial for tumorigenesis, such as cell motility and migration, negative regulation of cell death, angiogenesis, stem cell differentiation, and neuron development and differentiation were similarly present in all the samples (Appendix A). To further investigate the differences between the proteome content of GBM- and CSC-sEVs, we performed hierarchical clustering analysis of the identified proteins and found that the sEVs from the two U87 and U373 GBM cell lines clustered together and apart from the CSC-sEVs (Figure 6A). We then focused on three protein clusters (namely clusters 1, 7, and 8) whose protein profiles were clearly different between GBM- and CSC-sEVs (Figure 6A and Appendix A).

Then, to highlight the biological pathways which are unique or in common between GBM cell lines and CSCs, we selected the proteins involved in the biological functions of interest and present in clusters 1, 7, and 8 (Figure 6A). This process filters out 307 proteins (Appendix A), which are represented in the network graph shown in Figure 6B. Interestingly, we found that: (i) CSC-sEVs transport many more proteins related to RNA processing and gene expression functions with respect to GBM-sEVs; (ii) the majority of common proteins are most abundant in CSC-sEVs; (iii) the regulation of stem cell differentiation is conveyed in all cell lines; (iv) the regulation of cell adhesion (including cell junction and cell junction organization) is more represented in the GBM-sEVs, while proteins related to cell adhesion, angiogenesis, and regulation of migration are more present in CSC-sEVs. However, in the “regulation of migration” process, several proteins are more abundant in GBM-sEVs. Protein hubs mainly connected to all the biological functions of interest are represented in the middle of the network graph and include a set of proteins involved in the vesicle-mediated protein transport, most of which are involved in the endocytosis pathway, and the proteins catenin beta-1 (CTNNB1), RSP27A and neurofibromatosis type 1 (NF1).

Finally, we combined and compared biological processes enriched by miRNA target genes and proteins, to highlight functions that are commonly represented or uniquely carried between GBM- and CSC-sEVs and between miRNA targets and proteins. Interestingly, we found that the functions related to cell proliferation, cell death, and stemness are in common between GBM- and CSC-sEVs and are identified by both miRNA targets and proteins found in their sEVs (Figure 7 and Appendix A). While the functions of adhesion, migration, and angiogenesis, which we previously found associated with miRNA targets of only GBM-sEVs, resulted enriched also at protein level for both GBM- and CSC-sEVs, the EMT process showed enrichment only for miRNA targets of GBM-sEVs, with no evidence of protein involvement. Conversely, the functions related to membrane docking, cell junction organization, and endocytosis were enriched by proteins of both GBM- and CSC-sEVs, with no evidence (if not minimal for endocytosis) of miRNA target involvement. Lastly, the ncRNA-processing function was associated only with proteins of CSC-sEVs (Appendix A). Notably, the seven proteins described above with RNA-modifying activity regulating gene expression and found expressed exclusively in our CSC-sEVs (ADAR, METTL1, NAT10, PUS7, TRM112, TRM6, and TRMT61A; see Table 3) are included in the ncRNA-processing function (Appendix A).

## 4. Discussion

GBM is known as the most aggressive and malignant primary human brain tumor, with few diagnostic and treatment options. A subset of glioma tumor cells has been found to have stem cell-like characteristics. This cell population, named cancer stem cells (CSCs), also known as tumor-initiating cells, has been found to be highly oncogenic and plays a major role in the maintenance of GBM growth. When CSCs are transplanted into mice, the resulting tumors have the same features as the original tumor, and in vitro and in vivo investigations have shown that CSCs have multilineage differentiation capacity [57,58]. CSCs are able of acquiring both an epithelial/proliferating and a mesenchymal/invasive phenotype, showing great plasticity to switch between epithelial and mesenchymal phenotypes, and thus playing crucial roles in the GBM EMT, which contributes to the development of GBM chemo- and radiotherapy resistance [59]. In addition, CSCs can evade the immune system, promoting M2 polarization of resident macrophages and inhibiting T cell response [60,61].

It is now ascertained that sEVs promote tumor formation and progression by mediating the intercellular transport of miRNAs, mRNA, and proteins, and there is growing interest in their use as biomarkers for diagnosis and monitoring of disease recurrence [62]. In particular, sEVs released by GBM cells can stimulate their own malignancy by suppressing the immune response or affecting the tumor microenvironment [63]. Moreover, they can cross the BBB and carry their molecular cargo into peripheral circulation, thus laying the groundwork for the development of non-invasive diagnostic methods for this disease (the so-called “liquid biopsy”).

By defining the miRNA and protein content of the sEVs secreted by two functionally distinct cellular models (GBM cell lines and GBM primary CSCs), we aimed at providing data for better understanding the role of sEVs in these two cellular contexts. We chose the rapidly proliferating U87 MG and the slowly proliferating U373 MG GBM cell lines that were reported to show differences in gene expression and phenotype, resembling neuronal and mesenchymal characters, respectively [64]. The CSCs used in this study were derived in our lab from three human GBM tumors [5] and are characterized by strong proliferative capability in vitro and tumor-initiating ability. Studying CSCs in vitro may give us new insights and opportunities for targeting cell subpopulations that have previously been ignored when investigating GBM-established cell lines [65].

We defined the size and marker expression of sEVs isolated by ultracentrifugation, and our data are in agreement with the International Society for Extracellular Vesicles guidelines [10], according to which isolated EVs can be considered small EVs if, among other features, they are in the 30–200 nm size range and are enriched with exosomal markers, such as CD9, CD81, Alix, and TSG101, as we demonstrated using flow cytometry and MS.

Regarding the miRNome profiling, seven out of the 10 miRNAs most up-regulated in GBM-sEVs are known to display tumor suppressor functions, and no oncogenic miRNAs were found in this group (see Table 1 in green). In fact, according to our findings, GBM cell lines expand in culture faster than CSCs. We can hypothesize that cancer cell lines discard these tumor suppressor miRNAs to boost their in vitro proliferation and gain a survival advantage. Conversely, among the 10 most up-regulated miRNAs in CSC-sEVs, we found three miRNAs previously shown to have pro-tumor activity (miR-5189-3p, miR-3621, miR-135b-5p) and three others have tumor-suppression functions (miR-1236-5p, miR-363-3p, miR-450a-5p). CSC-sEVs could take advantage from expressing both tumor suppressor and oncogenic miRNAs, as these could facilitate an equilibrium in which stemness preservation of CSCs may allow them to escape pharmacological therapies. According to the miRTarBase database, the three pro-tumor miRNAs found in CSC-sEVs had 200 experimentally validated target genes that, based on our functional enrichment analysis, participate in the regulation of pluripotency and cell differentiation. Specifically, by targeting self-renewal genes, such as KLF4, SOX4, NOTCH1, and RPBJ, the CSCs—being more undifferentiated than GBM cell lines—have a lack of some functions, which are highlighted when they differentiate [4].

The MS analysis identified 249 proteins differentially expressed in GBM-sEVs as compared to CSC-sEVs. Among the proteins over-expressed in GBM-sEVs, we found tenascin C (TNC), an extracellular matrix molecule that drives the progression of many types of human cancer, including GBM [66]; CD109, which is associated with stemness maintenance and disease recurrence [67]; CD44, which has been implicated in malignant processes including cell motility, tumor growth, and angiogenesis in many cancer types and has been proposed as an invasion and migration marker in GBM [68]; and Ras suppressor protein 1 (RSU1), which is highly expressed in more aggressive GBM cells, promoting cell invasion [69]. Among the proteins over-expressed in CSC-sEVs, we underline the presence of MAP4K4, involved in the regulation of tumorigenesis and tumor progression [70]; N-Cadherin (CDH2), a signature of glioma stem cells [71]; Copine III (CPNE3), which regulates invasion, migration, and proliferation of GBM cells through the FAK pathway [72]; and Ephrin receptor type 2 (EPHA2), which is already known to be over-expressed in GBM CSCs, providing a measure of their stem-like potential and tumor-propagating ability, and to be involved in angiogenesis, migration, adhesion, proliferation, and differentiation [73].

Interestingly, we found that all sEVs secreted by GBM and CSCs contain proteins involved in mediating cell adhesion, such as ITGA, ITGB, and MCAM. It is likely that this class of proteins is useful to the sEVs for mediation of the docking at the cellular port in the recipient cells, thus bridging the entry of the sEV cargo from the cell surface into different target cells [74].

We found that RNA processing enzymes are selectively loaded into CSC-sEVs. rRNAs and tRNAs modifications enable genome decoding—especially in differentiating stem cells [75]—and in cancer, where they have been shown to promote proliferation [76] and drug-resistance [40]. Such modifications are catalyzed by dedicated enzymes both during homeostatic maturation and during cellular stress response [77]. mRNA modifications also appear to have functional roles in glioblastoma in controlling the expression of oncogenes and tumor suppressors [78,79]. Moreover, genomic studies revealed that the expression levels of selected RNA modification enzymes define the tumor microenvironment in glioblastoma [80]. We hypothesize that RNA modification enzymes retrieved in CSC-sEVs could promote preferential decoding of specific sets of genes to support stress response and survival. Similarly, the ribosomal proteins that are retrieved in CSC-sEVs (see Figure 6) could mediate the execution of translational programs, as shown in neuronal stem cells [81]. Future efforts will be necessary to address the question of whether those RNA modification enzymes and ribosomal proteins retain regulatory functions either in the extracellular vesicles or in the target cells.

Importantly, we generally found that the miRNA and protein contents of sEVs are not functionally correlated, meaning that sEVs do not incorporate miRNAs that target the same proteins that they contain. A key advantage of this molecular diversity of miRNAs and proteins packed in sEVs lies in providing recipient cells with several different regulatory options: (i) EV proteins can directly exert a biochemical effect into recipient cells upon release from sEVs, and (ii) miRNAs can suppress translation of proteins and/or degrade mRNA species in cells targeted by sEVs. Collectively, our observations suggest that sEVs shuttle a selective cargo of miRNAs and proteins, which is consistent with the important putative role of sEVs as vectors of cancer cell survival and intercellular communication signals. In addition, since sEVs are enriched with a molecular cargo carrying proteins and miRNAs with distinct functions, it is essential to link their function with regulatory pathways and biological activities that can be exploited for therapy. Our study shows that the miRNAs are linked to pathways that control transcription and presumably regulate gene expression in recipient cells. In contrast, proteins present in EVs appear to be involved in a broad spectrum of cellular signal transduction pathways that may control how recipient cells respond to external signals. As a result of our combined investigation of the miRNA and protein cargo of sEVs from GBM cell lines and CSCs, we suggest a paradigm in which sEVs can influence both the regulatory input and the phenotypic expression output of the cells with which they interact.

## Figures and Tables

**Figure 1 biomedicines-10-01886-f001:**
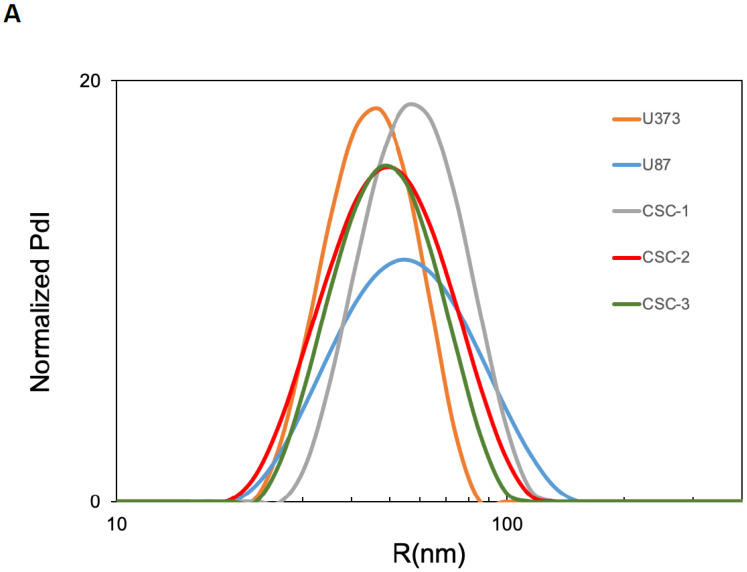
Characterization of sEVs released in cell culture media by U87, U373, and GBM primary CSCs. (**A**) Size of the released sEVs was measured via dynamic light scattering. The representative Intensity distribution curves are an average of five different measurements of the same sample. (**B**) sEVs purified from the different cell culture media were immunocaptured by magnetic Dynabeads conjugated with anti-CD63 tetraspanin antibodies, and bead-bound sEVs were processed for the detection of the indicated surface markers via immunofluorescence and flow cytometry. Aggregates and debris were excluded (gating) from the fluorescence analysis. In each cytogram, the reported number represents the percentage of positivity for the indicated marker. PdI, intensity distribution; SSC, side scatter; FL1, green fluorescence; FL4, far red fluorescence; PE, phycoerythrin; FITC, fluorescein isothiocyanate; ZONAB, ZO-1-associated Nucleic-Acid-Binding protein; GFAP, glial fibrillary acidic protein; POS, positive; NEG, negative.

**Figure 2 biomedicines-10-01886-f002:**
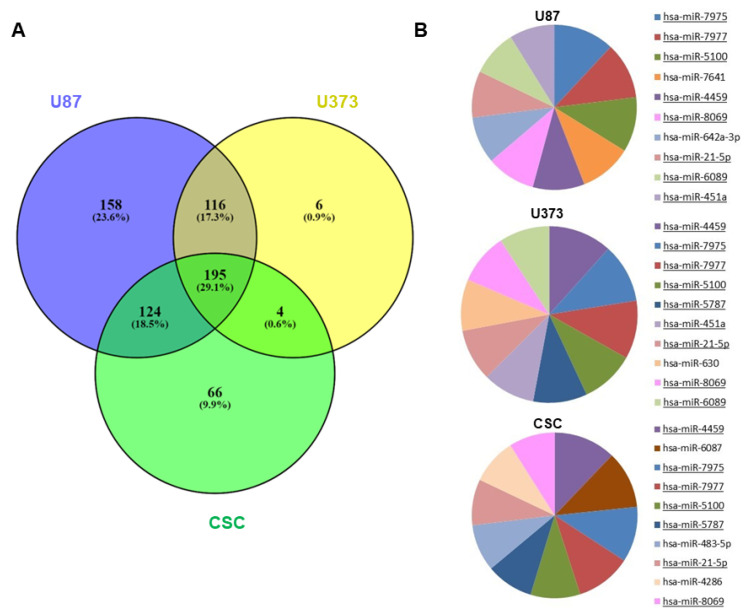
miRNA expression in GBM- and CSC-sEVs. (**A**) Venn diagram shows overlaps between the miRNAs expressed in sEVs secreted by U87, U373, and GBM primary CSCs. (**B**) The pie charts show the top 10 most abundant miRNAs in sEVs secreted by U87, U373, and GBM primary CSCs. MiRNAs common to two or more samples are underlined.

**Figure 3 biomedicines-10-01886-f003:**
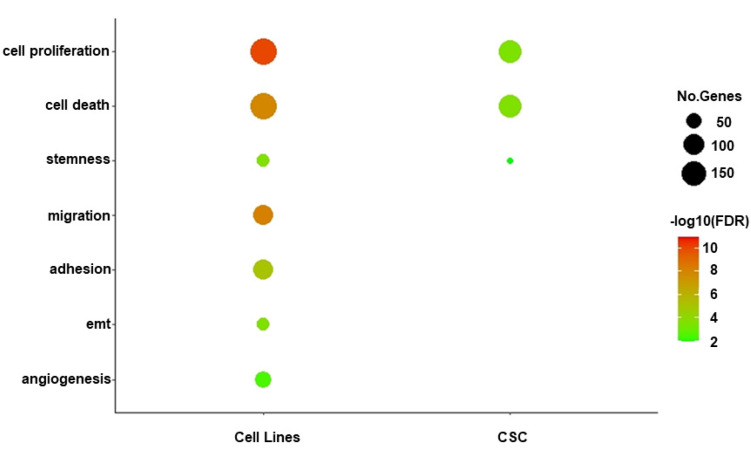
Functional enrichment analysis of miRNA target genes. Bubble plot shows a selection of biological functions of interest significantly enriched by miRNA target genes for GBM- and CSC-sEVs. Bubble size represents the number of target genes involved in each enriched function, while color gradient indicates statistical significance of the enrichment (FDR-BH, in log10 scale). Only significant enrichments are shown (FDR cutoff: 0.01).

**Figure 4 biomedicines-10-01886-f004:**
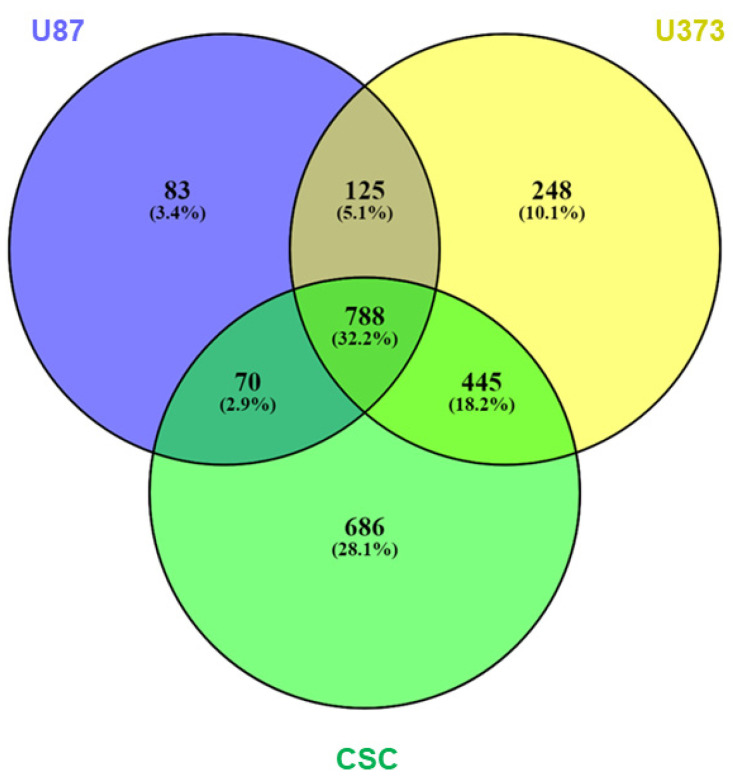
Protein expression in GBM- and CSC-sEVs. Venn diagram shows overlaps among the proteins expressed in sEVs secreted by U87, U373, and GBM primary CSCs. A total of 2445 proteins were identified and quantified, with at least two peptides in at least two biological replicates per sample: 1066 (U87), 1606 (U373), and 1989 (CSCs) proteins were carried in the respective secreted sEVs; 788 proteins were common to all the three samples.

**Figure 5 biomedicines-10-01886-f005:**
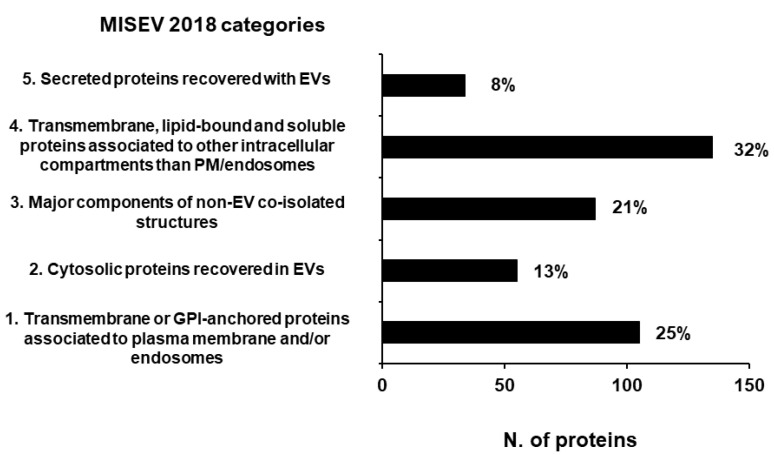
Proteins annotated in MISEV 2018 categories [10] assessing the quality of EV preparations. A total of 17% of our total protein dataset has been annotated in the MISEV2018 categories. The number reported next to each column is the relative percentage of the proteins annotated in that category.

**Figure 6 biomedicines-10-01886-f006:**
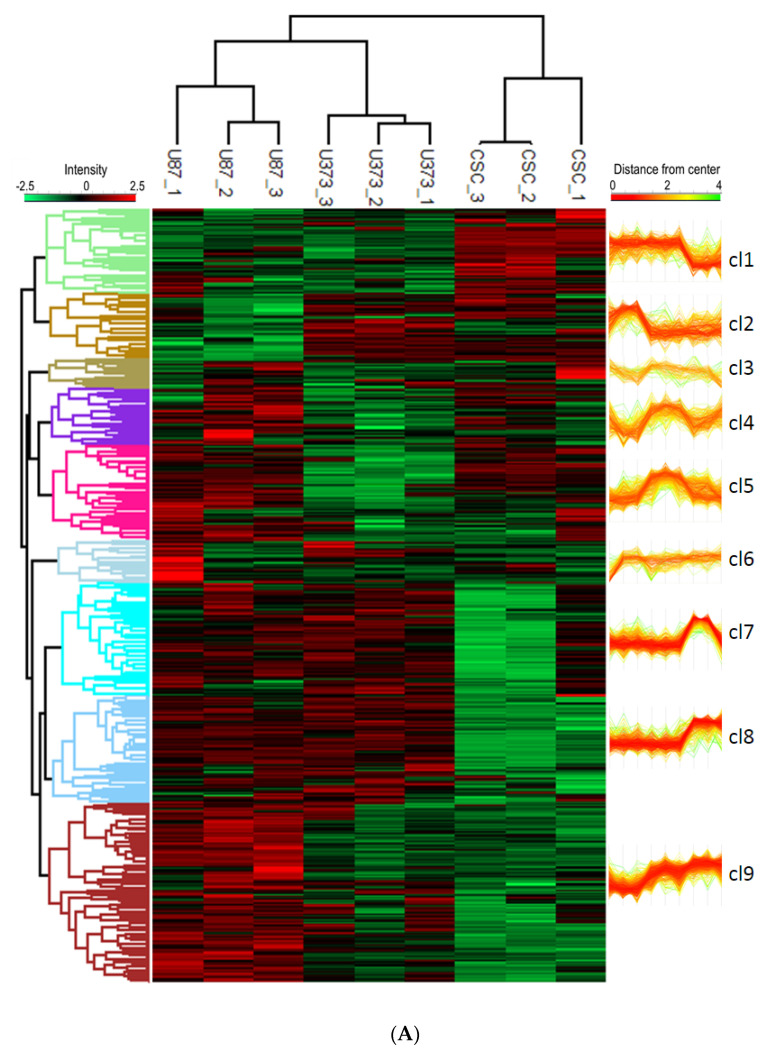
**(A**) Hierarchical clustering analysis of proteins in GBM- and CSC-sEVs. The clustering analysis is based on the Euclidean distance and complete linkage. The heatmap was divided into nine clusters (distance threshold = 4.5). (**B**) Functional enrichment analysis of proteins in GBM- and CSC-sEVs. In the String network, the degree of interconnection of each protein is proportional to the node size and the confidence of the interaction to the connecting edge size. The fill color scheme considers the presence of the protein in the samples, while the border color and thickness consider the significant quantitative difference between the CSC sample and established cell line samples (bottom right). The enriched GO Biological function term details are reported in Appendix A.

**Figure 7 biomedicines-10-01886-f007:**
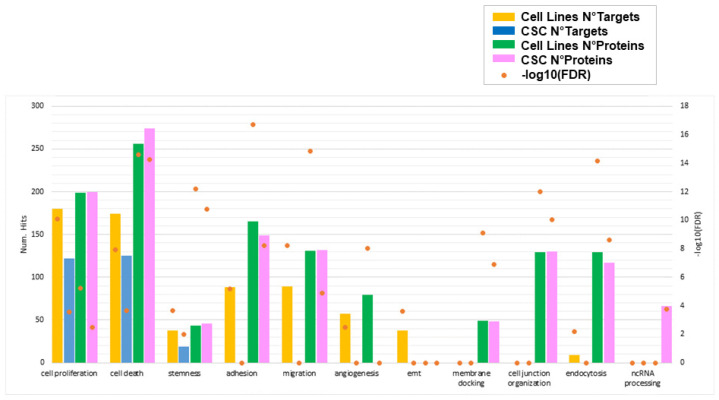
Biological functions enriched by miRNA target genes and/or proteins for sEVs secreted by GBM cell lines and primary CSCs. For each function of interest, the number of target genes and/or proteins enriching that process is plotted in different colors for GBM cell lines and CSCs. Orange points represent the statistical significance of the enrichments (FDR, in log10 scale). Only significant enrichments are shown (FDR cutoff: 0.01).

**Table 1 biomedicines-10-01886-t001:** Top 10 most up-regulated (top10) miRNAs in GBM-sEVs as compared to CSC-sEVs (upper) and top 10 miRNAs in CSC-sEVs as compared to GBM-sEVs (lower).

	Top10 Most Up-Regulated miRNAs
GBM-sEVs	hsa-miR-451a
	hsa-miR-4730
	hsa-miR-144-3p
	hsa-miR-6716-3p
	hsa-miR-451b
	hsa-miR-142-3p
	hsa-miR-6514-3p
	hsa-miR-3591-3p
	hsa-miR-126-3p
	hsa-miR-34b-3p
CSC-sEVs	hsa-miR-5189-3p
	hsa-miR-1236-5p
	hsa-miR-6769a-5p
	hsa-miR-3621
	hsa-miR-135b-5p
	hsa-miR-3937
	hsa-miR-363-3p
	hsa-miR-5572
	hsa-miR-450a-5p
	hsa-miR-8060

Top 10 miRNAs are listed according to decreasing fold-change values in GBM-sEVs vs. CSC-sEVs, and vice versa. Red, miRNAs with documented pro-tumor activity; green, miRNAs with documented tumor suppressor activity; black, miRNAs not described as cancer-related.

**Table 2 biomedicines-10-01886-t002:** Typical sEV markers identified in all our samples by MS *.

Mater Protein ID	Gene Name	Description	Exocarta Top100_EV #Reports	MISEV 2018 **
P31946	1433B	14-3-3 protein beta/alpha OS = Homo sapiens OX = 9606 GN = YWHAB PE = 1 SV = 3	258	2
P62258	1433E	14-3-3 protein epsilon OS = Homo sapiens OX = 9606 GN = YWHAE PE = 1 SV = 1	300	2
P61981	1433G	14-3-3 protein gamma OS = Homo sapiens OX = 9606 GN = YWHAG PE = 1 SV = 2	240	2
P27348	1433T	14-3-3 protein theta OS = Homo sapiens OX = 9606 GN = YWHAQ PE = 1 SV = 1	254	2
P63104	1433Z	14-3-3 protein zeta/delta OS = Homo sapiens OX = 9606 GN = YWHAZ PE = 1 SV = 1	301	2
P50995	ANXA11	Annexin A11 OS = Homo sapiens OX = 9606 GN = ANXA11 PE = 1 SV = 1	209	2
P04083	ANXA1	Annexin A1 OS = Homo sapiens OX = 9606 GN = ANXA1 PE = 1 SV = 2	251	2
P07355	ANXA2	Annexin A2 OS = Homo sapiens OX = 9606 GN = ANXA2 PE = 1 SV = 2	337	2
P08758	ANXA5	Annexin A5 OS = Homo sapiens OX = 9606 GN = ANXA5 PE = 1 SV = 2	313	2
P08133	ANXA6	Annexin A6 OS = Homo sapiens OX = 9606 GN = ANXA6 PE = 1 SV = 3	260	2
P20073	ANXA7	Annexin A7 OS = Homo sapiens OX = 9606 GN = ANXA7 PE = 1 SV = 3	228	2
P35613	BASI	Basigin OS = Homo sapiens OX = 9606 GN = BSG PE = 1 SV = 2	230	1
P60033	CD81	CD81 antigen OS = Homo sapiens OX = 9606 GN = CD81 PE = 1 SV = 1	262	1
P21926	CD9	CD9 antigen OS = Homo sapiens OX = 9606 GN = CD9 PE = 1 SV = 4	328	1
P15311	EZRI	Ezrin OS = Homo sapiens OX = 9606 GN = EZR PE = 1 SV = 4	262	2
P02751	FINC	Fibronectin OS = Homo sapiens OX = 9606 GN = FN1 PE = 1 SV = 5	233	5
O75955	FLOT1	Flotillin-1 OS = Homo sapiens OX = 9606 GN = FLOT1 PE = 1 SV = 3	259	2
P04406	G3P	Glyceraldehyde-3-phosphate dehydrogenase OS = Homo sapiens OX = 9606 GN = GAPDH PE = 1 SV = 3	377	2
P62873	GBB1	Guanine nucleotide-binding protein G(I)/G(S)/G(T) subunit beta-1 OS = Homo sapiens OX = 9606 GN = GNB1 PE = 1 SV = 3	257	1
P62879	GBB2	Guanine nucleotide-binding protein G(I)/G(S)/G(T) subunit beta-2 OS = Homo sapiens OX = 9606 GN = GNB2 PE = 1 SV = 3	240	1
P04899	GNAI2	Guanine nucleotide-binding protein G(i) subunit alpha-2 OS = Homo sapiens OX = 9606 GN = GNAI2 PE = 1 SV = 3	252	1
Q5JWF2	GNAS1	Guanine nucleotide-binding protein G(s) subunit alpha isoforms XLas OS = Homo sapiens OX = 9606 GN = GNAS PE = 1 SV = 2	226	1
P07900	HS90A	Heat shock protein HSP 90-alpha OS = Homo sapiens OX = 9606 GN = HSP90AA1 PE = 1 SV = 5	327	2
P08238	HS90B	Heat shock protein HSP 90-beta OS = Homo sapiens OX = 9606 GN = HSP90AB1 PE = 1 SV = 4	306	2
P11142	HSP7C	Heat shock cognate 71 kDa protein OS = Homo sapiens OX = 9606 GN = HSPA8 PE = 1 SV = 1	363	2
P05556	ITGB1	Integrin beta-1 OS = Homo sapiens OX = 9606 GN = ITGB1 PE = 1 SV = 2	250	1
Q08380	LG3BP	Galectin-3-binding protein OS = Homo sapiens OX = 9606 GN = LGALS3BP PE = 1 SV = 1	219	5
P26038	MOES	Moesin OS = Homo sapiens OX = 9606 GN = MSN PE = 1 SV = 3	266	1
Q8WUM4	PDC6I	Programmed cell death 6-interacting protein OS = Homo sapiens OX = 9606 GN = PDCD6IP PE = 1 SV = 1	399	2
P61026	RAB10	Ras-related protein Rab-10 OS = Homo sapiens OX = 9606 GN = RAB10 PE = 1 SV = 1	236	1
P51149	RAB7A	Ras-related protein Rab-7a OS = Homo sapiens OX = 9606 GN = RAB7A PE = 1 SV = 1	217	1
P61586	RHOA	Transforming protein RhoA OS = Homo sapiens OX = 9606 GN = RHOA PE = 1 SV = 1	220	2
O00560	SDCB1	Syntenin-1 OS = Homo sapiens OX = 9606 GN = SDCBP PE = 1 SV = 1	277	2
P68366	TBA4A	Tubulin alpha-4A chain OS = Homo sapiens OX = 9606 GN = TUBA4A PE = 1 SV = 1	216	2
P02786	TFR1	Transferrin receptor protein 1 OS = Homo sapiens OX = 9606 GN = TFRC PE = 1 SV = 2	211	1
Q99816	TSG101	Tumor susceptibility gene 101 protein OS = Homo sapiens OX = 9606 GN = TSG101 PE = 1 SV = 2	255	2

* Table shows proteins found in all our samples and matching both ExoCarta Top100 reports (http://www.exocarta.org/, accessed on 28 June 2022) and MISEV 2018 guidelines [10]. ** Numbers indicate MISEV categories for EV marker classification.

**Table 3 biomedicines-10-01886-t003:** RNA-modifying enzymes found in our GBM- and CSC-sEVs.

Name	Origin of EVs	Molecular Targets	Function	Involvement in Cancer	Refs.
ADAR1	CSCs	Editing of 3′-UTR GM2A ganglioside is linked to CSC self-renewal. Attenuated editing of miRNA-376a promotes GBM invasion. Editing independent-binding activity on CDK2 mRNA promotes proliferation of GBM cells	Adenosine-to-inosine RNA editing in physiology and cancer development. RNA editing independent-RNA binding activity	Its elevated expression correlates with poor prognosis in GBM	[38,39]
ELP1	U373	Elongator complex: promotes the mcm^5^s^2^ modification of the wobble uridine 34 of the tRNA anticodon	Ensure efficient translational decoding	The Elongator complex promotes resistance to targeted therapy	[40,41]
FBRL	U373, CSCs	Performs 2′–O–ribose methylation of rRNA 28S, in complex with box C/D snoRNAs	Ribosomal rRNA maturation	Expression of FBRL associates with poor prognosis in breast cancer	[42,43]
HCD2	U373, CSCs	Multifunctional tRNA processing enzyme. tRNA methylation (m1A, m1G at position 9) and pre-tRNA processing	Mitochondrial tRNA processing/mitochondrial fatty acid oxidation	Unexplored, but possible correlation with poor prognosis	[44,45]
METTL1	CSCs	Catalyzes m^7^G modification of tRNAs at position 46	Promotes tRNA stability	Drives oncogenic transformation, associates with poor survival in glioma	[46,47]
NAT10	CSCs	Catalyzes cytidine acetylation (ac4C) of the 28S rRNA. Catalyzes ac4C on tRNAs. Possible activity on mRNAs.	Translational efficiency	High expression correlates with poor prognosis, but also with tumor infiltrates (pan-cancer)	[48,49]
PUS7	CSCs	Pseudouridine synthetase, targets position 13 and possibly 8 of tRNAs, targets snRNAs, rRNAs and mRNAs	Regulates the biogenesis of tRNA fragments, regulates translational fidelity	High expression in glioma samples correlates with poor survival	[50,51]
TRM112	CSCs	Co-factor supporting the methyltransferase activity of various enzymes targeting tRNAs, rRNAs, and DNA	Translational regulation	Might depend on specific activities of binding partners	[52,53]
TRM6	CSCs	m1A- methylatransferase targeting position 58 of the initiator tRNA methionine tRNAi(Met). Functions in complex with TRMT61	Stabilization of tRNAi(Met)	Its elevated expression correlates with poor prognosis in glioma, might act as an oncogene by supporting translation of key transcripts	[54,55]
TRMT61A	CSCs	m1A- methylatransferase targeting position 58 of the initiator tRNA methionine tRNAi(Met). Functions in complex with TRMT61	Stabilization of tRNAi(Met)	Its elevated expression correlates with poor prognosis in glioma, might act as an oncogene by supporting translation of key transcripts	[56]

Shaded rows identify genes whose role in glioma is supported by published data. CSCs: cancer stem cells, mcm^5^s^2^: 5-methoxy-carbonyl-methyl-2-thio-uridine (mcm5s2U), m1A: N1-methyladenosine, m1G: N1-methylguanosine, m7G: N7-methylaguanosine, ac4C: N4-acetylcytidine, tRNA: transfer RNA, rRNA: ribosomal RNA, snoRNA: small nucleolar RNAs, miRNA: microRNA.

## Data Availability

miRNA raw data have been submitted to the GEO repository, under the Accession number GSE206675. Protein raw data have been uploaded to MassIVe and Proteo-meXchange repositories, under Accession numbers MSV000088671 and PXD030894.

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
