# Peer review of "miRNome and Proteome Profiling of Small Extracellular Vesicles Secreted by Human Glioblastoma Cell Lines and Primary Cancer Stem Cells"

_biomedicines, 2022, doi:10.3390/biomedicines10081886_

Round 1

Reviewer 1 Report

This peer-review is on the manuscript “miRNome and proteome profiling of small extracellular vesicles secreted by human glioblastoma cell lines and primary cancer stem cells” by Cifola and colleagues. The authors investigate and compare small EVs at the protein and microRNA levels from two established glioblastoma cell lines and three Cancer Stem Cell (CSC), i.e. short term cultured neurosphere lines from glioblastoma patients.

Although the manuscript is well written and the many data appear to be solid, there are some aspects which should be addressed in more detail:

1)     The number of 2 cell lines and 3 CSC lines is rather low to make comparisons at all, but also to generalize between established cell lines and CSCs, especially when pooling 3 different tumors together. It should be mentioned why it was necessary to pool the supernatants from three different CSCs, i.e. from three different patients. It is not clear, if (or why not) a mutational profile of the primary tumors was available – and how they are comparable from mutational status with the two selected cell lines and different subtypes of glioblastoma. For example, the cell lines may have p53 mutations, the primary tumors not; or different mutational profiles in terms of PTEN, LOH chromosome 10, EGFR, and many more widely known mutations which may have a major impact on sEVs. The authors should mention some of these limitations, perhaps in the discussion.

2)     Line 107-109: “In the present study, we investigated the miRNome and proteome of the small EVs  (sEVs) released by two different human GBM cell lines (GBM) and by three GBM primary cancer stem cells (CSCs).” Perhaps rephrase, since not three cells were analyzed, but tumor-derived neurospheres with many cells from three different patients. Perhaps mentioning here the difference of (short-term cultured) CSCs to the established cell lines may help the interested reader (but it may be OK to keep it in the materials/methods section).

3)     l. 355: “Among the top10, all but one (hsa-miR-363-3p) were miRNAs exclusively expressed in GBM-sEVs or CSC-sEVs, showing that these miRNAs distinguish the two sample types and may possibly target separate functions.” The list in Table 1 doesn’t show hsa-miR-363-3p in the GBM-sEVs, the discrepance should be explained (averaging error? Supplement has it in one established cell line), or the Table or text corrected.

4)     Figure 6. (A) CSC1, 2, 3, but Figure 1 refers to CSC2, 3, 4. Please check all Figures and correct discrepancy.

5)     The authors found that miRNA of sEVs from CSCs can contain both pro-tumor activity and tumor suppressor activity. The authors may assume that the same sEV contains both, but this should be clarified or discussed in more detail. Could there be sEVs with pro-tumor and other sEVs with tumor-supressor markers? Similar aspect on GBM-sEVs (established cell lines) – do the experiments suggest that all top10 miRNAs are present in each sEV, or is it likely that varying differentiation states of cells produce different sEVs?

Author Response

Comments and Suggestions for Authors

This peer-review is on the manuscript “miRNome and proteome profiling of small extracellular vesicles secreted by human glioblastoma cell lines and primary cancer stem cells” by Cifola and colleagues. The authors investigate and compare small EVs at the protein and microRNA levels from two established glioblastoma cell lines and three Cancer Stem Cell (CSC), i.e. short term cultured neurosphere lines from glioblastoma patients.

Although the manuscript is well written and the many data appear to be solid, there are some aspects which should be addressed in more detail:

1)     The number of 2 cell lines and 3 CSC lines is rather low to make comparisons at all, but also to generalize between established cell lines and CSCs, especially when pooling 3 different tumors together. It should be mentioned why it was necessary to pool the supernatants from three different CSCs, i.e. from three different patients. It is not clear, if (or why not) a mutational profile of the primary tumors was available – and how they are comparable from mutational status with the two selected cell lines and different subtypes of glioblastoma. For example, the cell lines may have p53 mutations, the primary tumors not; or different mutational profiles in terms of PTEN, LOH chromosome 10, EGFR, and many more widely known mutations which may have a major impact on sEVs. The authors should mention some of these limitations, perhaps in the discussion.

A.: Following the reviewer’s suggestion, we better explained in the section 3.2 why we pooled the supernatants from the three CSC cultures. Indeed, due to their relatively slow proliferation rate, the three CSC lines released low amounts of sEVs and it was necessary to pool the supernatants from the three cultures to have sEV amounts comparable to those obtained from the two GBM cell lines.

Concerning the analysis of the mutational status of the CSCs with respect to the GBM cell lines, we greatly appreciate the reviewer’s comment. Indeed, we have ongoing an exome sequencing study of several primary CSC cultures obtained from GBM patients, comparing the tumor core burden with the peripheral tissue, but we think that this could be the object of a future more detailed and focused work.

2)     Line 107-109: “In the present study, we investigated the miRNome and proteome of the small EVs  (sEVs) released by two different human GBM cell lines (GBM) and by three GBM primary cancer stem cells (CSCs).” Perhaps rephrase, since not three cells were analyzed, but tumor-derived neurospheres with many cells from three different patients. Perhaps mentioning here the difference of (short-term cultured) CSCs to the established cell lines may help the interested reader (but it may be OK to keep it in the materials/methods section).

A.: We apologize for our mistake and thank the reviewer. We rephrased the sentence according to her/his suggestion. We kept the description of the difference between short-term cultures and established lines in Methods section and in Discussion.

3)     l. 355: “Among the top10, all but one (hsa-miR-363-3p) were miRNAs exclusively expressed in GBM-sEVs or CSC-sEVs, showing that these miRNAs distinguish the two sample types and may possibly target separate functions.” The list in Table 1 doesn’t show hsa-miR-363-3p in the GBM-sEVs, the discrepance should be explained (averaging error? Supplement has it in one established cell line), or the Table or text corrected.

A.: In Table 1 we showed the top10 most up-regulated miRNAs in GBM-sEVs as compared to CSC-sEVs (upper part of the table), and the top10 most up-regulated miRNAs in CSC-sEVs as compared to GBM-sEVs (bottom part of the table). Hsa-miR-363-3p is one of the top10 most up-regulated miRNAs in CSC- vs GBM-sEVs and is listed in the bottom part of the table. It was found expressed at low levels also in U373-sEVs (while absent in U87-sEVs), but it is not included among the first 10 most up-regulated miRNAs in GBM- vs CSC-sEVs. For this reason, it is not listed in Table 1 in the upper part among the Top10 miRNAs of GBM-sEVs. Probably, we were not sufficiently clear when describing the results shown in Table 1. Thus, we rephrased the sentences (lines 344 to 357) and the title of Table 1.

4)     Figure 6. (A) CSC1, 2, 3, but Figure 1 refers to CSC2, 3, 4. Please check all Figures and correct discrepancy.

A.: We apologize for the mistake. The CSC numbering has been corrected in the new version of Figure 1.

5)     The authors found that miRNA of sEVs from CSCs can contain both pro-tumor activity and tumor suppressor activity. The authors may assume that the same sEV contains both, but this should be clarified or discussed in more detail. Could there be sEVs with pro-tumor and other sEVs with tumor-supressor markers? Similar aspect on GBM-sEVs (established cell lines) – do the experiments suggest that all top10 miRNAs are present in each sEV, or is it likely that varying differentiation states of cells produce different sEVs?

A.: We thank the reviewer for this comment, it is a very good point. Indeed, we did not perform such experiments. To address it, experimental approaches that analyze EVs at the single-particle level (e.g., single-cell RNA-seq) would be necessary. A great difficulty is the nano size of the EVs, especially of the small EVs, which is the vesicle population we studied. Sorting of single little particles is difficult due to the resolution limits of most flow cytometers. However, much effort is being made to separate single extracellular vesicles by immunoaffinity methods. The principal criticisms are: i) the identification of a surface marker that is specific for selected population as small extracellular vesicles; ii) the difficulty to obtain single separated vesicles, since they are extremely “sticky”; iii) the possibility to perform sequencing on a very low amount of RNA such as that can be extracted from small extracellular vesicles. We think that this could be resolved in the next few years but at the moment it remains a very attracting project.

Reviewer 2 Report

By using 2 human glioblastoma cell lines and 1 primary cancer stem cells, as well as multiple analysis methods, authors identified the miRNA and proteins of small extracellular vesicles in Glioblastoma functions. Overall, the manuscript is well prepared. I only have some suggestions to improve the manuscript.

In the introduction, authors mentioned that “Numerous studies have been conducted to identify miRNA and protein signatures 101 of GBM-derived EVs, however, due to significant variations among these investigations, 102 these biomarkers have yet to be implemented in clinical practice.” Please explain whether and how authors solved this important issue in the current manuscript.

Please check the manuscript carefully and correct the mistakes and errors. For example: Line 52, 57, 126 (2 mM, CO2), line 141 (2 mM, CO2), 157 (cm2), 169 (300xg), 213 (100 ng, RNA), 250 (60 mM, pH 6.8), 252 (50 mM), 277, 282-285 (FDR < 0.05), 298 and so on.

Please try to change the color of Figure 1A. It is difficult to distinguish the groups.

The resolution of Figure 1B is pretty low and is difficult to see the labeling.

The resolution of Figure 6 is pretty low and is difficult to see the labeling.

Author Response

Comments and Suggestions for Authors

By using 2 human glioblastoma cell lines and 1 primary cancer stem cells, as well as multiple analysis methods, authors identified the miRNA and proteins of small extracellular vesicles in Glioblastoma functions. Overall, the manuscript is well prepared. I only have some suggestions to improve the manuscript.

In the introduction, authors mentioned that “Numerous studies have been conducted to identify miRNA and protein signatures of GBM-derived EVs, however, due to significant variations among these investigations, these biomarkers have yet to be implemented in clinical practice.” Please explain whether and how authors solved this important issue in the current manuscript.

A.: Indeed, we were not able to solve this huge issue in our manuscript. Nonetheless, we tried to move a little step further by combining two types of omics data generated on the same vesicles. In our opinion, this would allow to obtain a more comprehensive picture of the content of small EVs and might aid the identification of the more important functional messages played by them in the context of glioblastoma biology.

Please check the manuscript carefully and correct the mistakes and errors. For example: Line 52, 57, 126 (2 mM, CO2), line 141 (2 mM, CO2), 157 (cm2), 169 (300xg), 213 (100 ng, RNA), 250 (60 mM, pH 6.8), 252 (50 mM), 277, 282-285 (FDR < 0.05), 298 and so on.

A.: Typos, mistakes and errors have been corrected throughout the entire manuscript.

Please try to change the color of Figure 1A. It is difficult to distinguish the groups.

A.: Colors in Figure 1A have been changed to be more distinguishable.

The resolution of Figure 1B is pretty low and is difficult to see the labeling.

A.: We increased the resolution of Figure 1B to improve image readability.

The resolution of Figure 6 is pretty low and is difficult to see the labeling.

A.: We improved the resolution of Figure 6.